# Innovative Bi_5_O_7_I/MIL-101(Cr) Compounds: A Leap Forward in Photocatalytic Tetracycline Removal

**DOI:** 10.3390/ijms25126759

**Published:** 2024-06-19

**Authors:** Jie Hong, Zhaohan Chu, Claudia Li, Wanliang Yang, Sibudjing Kawi, Qinong Ye

**Affiliations:** 1GuiZhou University Medical College, Guiyang 550025, China; 2North Alabama International College of Engineering and Technology, Guizhou University, Guiyang 550025, China; chuzhaohan007@163.com; 3Department of Chemical and Biomolecular Engineering, National University of Singapore, 4 Engineering Drive 4, Singapore 119260, Singapore; cli@nus.edu.sg; 4School of Chemistry and Chemical Engineering, Guizhou University, Guiyang 550025, China; yangwanlianghhhh@163.com

**Keywords:** Bi_5_O_7_I, MIL-101(Cr), tetracycline oxidation, heterojunction, metal–organic frameworks, visible light photocatalysis

## Abstract

In environmental chemistry, photocatalysts for eliminating organic contaminants in water have gained significant interest. Our study introduces a unique heterostructure combining MIL-101(Cr) and bismuth oxyiodide (Bi_5_O_7_I). We evaluated this nanostructure’s efficiency in adsorbing and degrading tetracycline (TC) under visible light. The Bi_5_O_7_I@MIL-101(Cr) composite, with a surface area of 637 m^2^/g, prevents self-aggregation seen in its components, enhancing visible light absorption. Its photocatalytic efficiency surpassed Bi_5_O_7_I and MIL-101(Cr) by 33.4 and 9.2 times, respectively. Comprehensive analyses, including scanning electron microscopy (SEM) and transmission electron microscopy (TEM), confirmed the successful formation of the heterostructure with defined morphological characteristics. BET analysis demonstrated its high surface area, while X-ray diffraction (XRD) confirmed its crystallinity. Electron spin resonance (ESR) tests showed significant generation of reactive oxygen species (ROS) like h^+^ and·•O_2_^−^ under light, crucial for TC degradation. The material maintained exceptional durability over five cycles. Density functional theory (DFT) simulations and empirical investigations revealed a type I heterojunction between Bi_5_O_7_I and MIL-101(Cr), facilitating efficient electron–hole pair separation. This study underscores the superior photocatalytic activity and stability of Bi_5_O_7_I@MIL-101(Cr), offering insights into designing innovative photocatalysts for water purification.

## 1. Introduction

The extensive use of tetracycline (TC) antibiotics in healthcare and agriculture has led to significant environmental pollution, especially in aquatic systems [1]. Increasing pollution levels contribute to the evolution of antibiotic-resistant bacteria, posing risks to both the ecosystem and public health [2]. Recent advances in research have highlighted the persistent properties of TCs, which cannot be completely removed from aquatic systems using traditional wastewater treatment methods. Although advanced treatment methods such as oxidation processes, adsorption methods, and ion exchange present potential for TC removal, they are hindered by high costs and operational challenges. Biodegradation appears to be a sustainable option, yet its practical application remains limited by efficiency and scalability issues [3]. Given the major environmental and health risks posed by antibiotic pollution, especially with the presence of antibiotic residues in aquatic environments, there is an urgent need to develop new, effective pollutant removal technologies [4,5,6]. In this context, photocatalytic technology has emerged as a cutting-edge research direction due to its ability to degrade pollutants using visible light. In the field of environmental chemistry, the development of photocatalysts for removing organic pollutants from water has been a key research direction [7]. This includes studies on the modification of polymer semiconductor-like graphitic carbon nitride (g-C_3_N_4_) and research on the photocatalytic degradation of TC using copper sulfide nanoparticles [8,9]. Although significant progress has been made, challenges in the stability of the catalyst and the efficiency of visible light absorption remain. Among these various semiconductor photocatalytic materials, heterojunctions have demonstrated potential in addressing this global issue due to their high electron–hole pair separation efficiency and excellent photocatalytic activity [2,3,4].

Bismuth oxyhalides (BiO_X_, where X = Cl, Br, I) have garnered widespread attention over the past decade due to their exceptional photocatalytic activity, driven by their layered structure and optimal bandgap for visible light absorption [10,11]. Compared to general bismuth compounds, the higher oxidation activity of bismuth oxyhalides is attributed to the elevated valence band position from the increased bismuth content, outperforming O_2_/H_2_O [12,13]. Moreover, bismuth oxyiodide (Bi_5_O_7_I) has more positive valence band (VB) edges than other Bi_x_O_y_X_z_ materials, implying its ability to generate more reactive species (H^+^ and/or •OH) to participate in the oxidation of organic pollutants [14]. Although Bi_5_O_7_I already exhibits good photocatalytic performance, its application is limited by the rapid recombination rate of photogenerated carriers and photocorrosion phenomena due to its wide bandgap. To address this, recent studies have optimized the photocatalytic performance of Bi_5_O_7_I, such as through the enhancement of the spatial separation efficiency of photogenerated electron–hole pairs through the construction of binary heterostructures with an internal electric field at the interface between different photocatalysts as reported by Huang et al. [14]. Tan et al. [15] studied the preparation of heterojunction composites with Bi_5_O_7_I by identifying suitable photocatalysts as the second component as an effective method to improve its photocatalytic activity. Additionally, employing NH_2_-UIO-66 as a co-catalyst to prepare Bi_5_O_7_I/UIO-66-NH_2_ heterojunction composites has enhanced the photocatalytic activity for ciprofloxacin (CIP) degradation [16].

MIL-101(Cr), which is part of the renowned Materials of the Institute Lavoisier series, stands out as a highly studied metal–organic framework (MOF). Its distinct structure, composed of Cr_3_O clusters interconnected by terephthalic acid, mirrors the MTN zeolite topology [16]. MIL-101(Cr) is distinguished by its mesoporous cages, available in two dimensions (29 Å and 34 Å), with pore openings reaching up to 16 Å, coupled with a Brunauer–Emmet–Teller (BET) specific surface area of 4100 m^2^/g. The deliberate removal of crystalline water molecules under specific conditions grants MIL-101(Cr) exceptional porosity and exposes unsaturated metal sites, thereby amplifying its utility across various fields such as electrocatalysis, photocatalysis, pollutant adsorption, and drug delivery, among others. The expanding research on MIL-101(Cr) underscores its crucial role in enhancing the application of MOFs in diverse sectors, thus contributing significantly to environmental conservation, energy storage, and the medical field [17]. The growing body of research around MIL-101(Cr) emphasizes its significance in advancing the application of MOFs across various domains, reflecting the broader impact of MOFs on environmental sustainability, energy storage, and healthcare [18]. However, research into the combination of MIL-101(Cr) and Bi_5_O_7_I remains unexplored, making our innovative synthesis of their heterostructure a pioneering endeavor to investigate its application in photocatalysis.

In this study, we developed a unique heterostructure, namely Bi_5_O_7_I@MIL-101, featuring a composite photocatalyst that exhibits an extraordinarily high specific surface area alongside suitable active sites. The innovative design serves to improve the interaction between the photocatalyst and pollutants via enhanced adsorption capacity and to expedite the rates of charge transfer compared to single component-based photocatalysts. The heterostructure demonstrated effective photocatalytic degradation of TC from water. The experimental findings revealed that the photocatalytic efficiency of the newly prepared Bi_5_O_7_I@MIL-101 composite material far surpassed that of both Bi_5_O_7_I and MIL-101(Cr) when used independently. The exceptional photocatalytic efficacy of the Bi_5_O_7_I@MIL-101 composite is owed to the type I heterojunction established at the 3D contact interface between the MIL-101(Cr) MOF and the Bi_5_O_7_I nanorods. This results in significant and robust absorption of visible light and effective separation of photogenerated carriers.

## 2. Results and Discussion

### 2.1. Material Characterizations

As shown in Figure 1a–c, scanning electron microscopy (SEM) was used to analyze the morphology of pure Bi_5_O_7_I, MIL-101(Cr), and Bi_5_O_7_I@MIL-101. The submicron size and uniformity of the three materials are clearly visible. Pure Bi_5_O_7_I exhibits irregular sizes and uneven lengths, forming large (micron-sized) self-assembled nanorod aggregates. These nanorods are densely packed in a disordered manner, severely limiting their adsorption, dispersion properties, and light absorption capacity (Figure 1a). On the other hand, Figure 1b shows that MIL-101(Cr) is composed of regular nanocrystal aggregates of small cubic particles, whose morphology also limits their dispersibility, thus restricting their catalytic performance. Figure 1c demonstrates that during the synthesis process, Bi_5_O_7_I transforms into smaller fibrous structures that are adhered to smaller MIL-101(Cr) crystals. MIL-101(Cr) is successfully dispersed and encapsulated, resulting in a rough nanorod structure. This significantly increases the specific surface area and light absorption capacity of the material, markedly reducing agglomeration and enhancing the specific surface area of the nanocrystals, thereby significantly improving the photocatalytic performance. Figure 1h,i clearly show smaller MOF particles attached to thinner Bi_5_O_7_I fibrous rods, consistent with the results in Figure 1c, confirming the formation of our composite. Figure 1d–g are elemental mapping images of Bi_5_O_7_I@MIL-101 (Figure 1c). Figure 1d shows the distribution of O elements; since both Bi_5_O_7_I and MIL-101 contain O elements, we can see that the synthesized catalyst is uniformly mixed. Figure 1e shows the Cr element, which is concentrated on the MIL-101(Cr) crystals on the fibrous threads. Figure 1f,g clearly show fibrous traces, proving that they are Bi_5_O_7_I. The elemental mapping images confirm the successful synthesis of our heterojunction, consistent with the TEM results. Figure 1j,k show the EDS mass content, with O at 40%, Cr at 6%, I at 4%, and Bi at 50%, demonstrating that the composite contains both MIL-101(Cr) and Bi_5_O_7_I. In summary, it is proved that a Bi_5_O_7_I@MIL-101(Cr) heterojunction is formed.

For a deeper examination of the microstructures of the photocatalysts, the nitrogen adsorption–desorption isotherms were analyzed. As illustrated in Figure 2, Bi_5_O_7_I@MIL-101(Cr) demonstrated a remarkable adsorption capacity at lower relative pressures, highlighting its well-developed microporous structure [19]. Bi_5_O_7_I exhibited a gradual increase in adsorption with rising relative pressure, yet it did not reach a plateau, indicating complex pore structures or less than ideal adsorption scenarios. Meanwhile, MIL-101(Cr) and Bi_5_O_7_I@MIL-101(Cr) displayed a more complex adsorption behavior. At lower relative pressures (p/P_0_ < 0.2), their adsorption capacity quickly increased and tended towards saturation, primarily due to the filling of micropores. Beyond a relative pressure of 0.2, a plateau emerged whereby the adsorption capacity increased slowly, ultimately approaching atmospheric pressure. This behavior may suggest a mixture or irregular pore structures, likely indicative of type I isotherms, which are typically associated with microporous materials [20]. The specific surface area of Bi_5_O_7_I@MIL-101(Cr), calculated to be 637 m^2^/g, notably exceeded those of MIL-101(Cr) (577 m^2^/g) and pure Bi_5_O_7_I (5 m^2^/g). According to BJH analysis, the composite material of Bi_5_O_7_I@MIL-101(Cr) was found to have a microporous framework, with the majority of pore diameters centered at approximately 4.26 nm and an overall pore volume of 0.33 cm^3^/g. The reduction in the average pore size of Bi_5_O_7_I@MIL-101(Cr), when compared to the individual components, suggests an interaction between the nanoparticles of MIL-101(Cr) and the layers of Bi_5_O_7_I, which likely led to a decrease in pore size. Generally, a larger surface area provides more sites for catalysis, thereby enhancing the photocatalytic performance [21].

X-ray diffraction (XRD) analysis was conducted to examine the structural details and crystallinity of the samples. As shown in Figure 3, all samples demonstrated high crystallinity. The XRD spectrum of Bi_5_O_7_I, with peaks at 2θ = 9°, 29.7°, 31.7°, 45.4°, and 55.2°, corresponded precisely with the Bi_5_O_7_I standard (JCPDS No. 40-0548) [22]. MIL-101(Cr) exhibited diffraction peaks akin to those of PDF#89-2697, particularly at 18.7° and 21.8° [23]. The pronounced diffraction peaks for both Bi_5_O_7_I and MIL-101(Cr) indicated their exceptional crystallinity, reflecting a high degree of structural order in the synthesized products. In the XRD pattern of the Bi_5_O_7_I@MIL-101(Cr) composite material, the peak at 18.7° associated with the MOF disappeared. This is attributed to the regrowth of the structure in the presence of Bi_5_O_7_I during synthesis, resulting in a modified structure where the main MOF peak remains but the 18.7° peak vanishes. This new structure also leads to an increase in specific surface area, consistent with the BET test results. Additionally, the XRD analysis shows that the composite retains the main peaks of MOF, while also displaying peaks at 9°, 45.4°, and 55.2° corresponding to Bi_5_O_7_I. The intensity of the diffraction peaks is significantly reduced, indicating that the Bi_5_O_7_I content was reduced. This confirms the presence of both Bi_5_O_7_I and MOF in the composite, which aligns with the SEM and TEM results, further verifying the formation of the Bi_5_O_7_I@MIL-101(Cr) heterojunction.

In the FTIR spectrum (Figure 4a), the weaker, sharper bands at 1011 and 752 cm^−1^ correspond to the δ(C-H) and γ(C-H) vibrations in aromatic structures [24]. The peak at 619 cm^−1^ is attributed to the in-plane bending vibrations of Cr-O groups [25]. On the other hand, the Bi_5_O_7_I FTIR spectra display distinct bands around 3500 cm^−1^ and 1600 cm^−1^. These bands are attributed to the symmetric stretching vibrations of OH groups and the bending vibrations of adsorbed water molecules, respectively. Additionally, absorption bands at 512 and 770 cm^−1^ are associated with Bi-O-O stretching vibrations, while those between 512 and 1420 cm^−1^ correspond to the stretching vibrations within the tetragonal crystal lattice of Bi_5_O_7_I [26,27]. In the FTIR spectrum of the composite Bi_5_O_7_I@MIL-101, these characteristic peaks were retained, suggesting MIL-101(Cr) retained its structural integrity in the heterostructure formation. In addition, the absorption peaks between 1600 and 1650 cm^−1^ of MIL-101(Cr) in FTIR represent the C=C vibrations in the aromatic rings and the C=O stretching in the dicarboxylate groups, which verify the peaks depicted in the Raman spectra (Figure 4b) [28]. The Raman bands at 1400 and 1625 cm^−1^ belong to the organic linker in the MOF. The peak at 1625 cm⁻^1^ may be related to the vibrations of C=C bonds, which can occur in many organic compounds and MOF structures. The above results are consistent with the SEM and XRD analysis results, further confirming the formation of the heterojunction between MOF and Bi_5_O_7_I.

As shown in Figure 5a–e, the surface composition and valence states of Bi_5_O_7_I@MIL-101 were characterized using X-ray photoelectron spectroscopy (XPS). Figure 5a indicates the presence of I, O, Cr, and Bi in the measured spectrum, further supporting the presence of both Bi_5_O_7_I and MIL-101(Cr) in the composite. The high-resolution spectrum of Bi in Figure 5b can be deconvoluted into appropriate peaks around 159 eV and 165 eV [29,30]. In the O 1s peak XPS spectrum (Figure 5c), the peak near 530 eV is attributed to the Bi–O bond. I 3d can be fitted with two characteristic peaks in Figure 5d. The two main peaks are located at 619.7 eV and 631.8 eV, respectively, assigned to I3 d_5/2_ and I 3d_3/2_ [31]. It can be proved that the complex exists in the form of Bi_5_O_7_I. Peaks at 577.1 and 587.0 eV correspond to Cr 2p_3/2_ and Cr 2p_1/2_, respectively. The Cr element is present in the composite, and its form is consistent with that in MIL-101(Cr) [32].Overall, the XPS analysis results further confirm the presence of Bi, O, I, and Cr in Bi_5_O_7_I@MIL-101(Cr). The presence and corresponding valence states of these elements are consistent with the XRD and FTIR results, indicating the successful formation of the Bi_5_O_7_I@MIL-101(Cr) composite.

Heterojunction structures typically exhibit superior photocatalytic performance compared to their individual components. This is primarily due to two key factors: enhanced light harvesting and improved electron–hole separation. In this study, the optical properties of Bi_5_O_7_I, MIL-101, and Bi_5_O_7_I@MIL-101 were investigated using UV-Vis. As shown in Figure 6a, the UV-Vis spectra of the samples indicate that Bi_5_O_7_I hardly absorbs visible light, while MIL-101(Cr) absorbs in the range of 200–800 nm. Notably, the composite material absorbs almost the entire wavelength spectrum after the formation of the heterojunction. This broadened absorption range enhances the generation of photoinduced charge carriers. This red shift phenomenon can be attributed to the incorporation of MIL-101(Cr), which not only provides more active sites but also promotes the effective separation of photogenerated electrons and holes. XRD and SEM analyses reveal that the Bi_5_O_7_I@MIL-101 composite has a larger specific surface area and optimized crystal structure, contributing to its improved photocatalytic performance. Compared to the individual materials, the composite demonstrates higher photocatalytic degradation efficiency of TC, further validating the superiority of the heterojunction structure in photocatalytic applications. In summary, the UV-Vis spectral analysis results are consistent with the XPS, XRD, and SEM analyses, confirming the successful synthesis of the Bi_5_O_7_I@MIL-101 heterojunction composite and its potential advantages in photocatalytic applications. Furthermore, the band edges of the materials were analyzed based on the Tauc plot, derived from the following formula:(1)$${\left(\alpha h\nu \right)}^{n}=A\left(h\nu -E_{g}\right)$$

Here, $E_{g}$, α, hν, and (A) represent the bandgap, absorption wavelength, absorption coefficient, photon energy, and a constant, respectively. The value of (n) depends on the transition characteristics of the semiconductor, which in this case is 1/2 [31,33]. The bandgaps of the photocatalysts are determined from the plot of ${\left(\alpha h\nu \right)}^{2}$ against the photon energy hν. The estimated bandgaps for Bi_5_O_7_I, MIL-101, and Bi_5_O_7_I@MIL-101 are 3.36 eV, 1.77 eV, and 1.70 eV, respectively (Figure 6b). The UV-Vis analysis indicates that the Bi_5_O_7_I@MIL-101 sample, upon forming a heterojunction, exhibits a narrowed bandgap with enhanced light absorption capabilities, thus facilitating the separation of photogenerated carriers and demonstrating higher photocatalytic performance.

The thermal stability and quantitative composition of the Bi_5_O_7_I@MIL-101 composite were analyzed using thermogravimetric analysis (TGA). As shown in Figure 7, the weight loss of the Bi_5_O_7_I@MIL-101 composite between 30 and 300 °C is 19.62 wt%, mainly due to the evaporation of physically and chemically adsorbed water. Significant weight loss occurs between 300 °C and 500 °C, attributed to the combustion of the organic components and the carbon framework of Bi_5_O_7_I and MIL-101(Cr) [34]. Upon heating beyond 500 °C, the residues primarily consist of B_i2_O_3_ and Cr_2_O_3_, which remain stable at the maximum temperature of 800 °C in the TGA analysis. This indicates that the Bi_5_O_7_I@MIL-101 composite has good thermal stability below 300 °C, making it suitable for practical applications. In addition, the SEM analysis (Figure 1c) reveals that the composite material exhibits a uniform surface morphology with a distinct porous structure. This structure is conducive to maintaining the catalyst’s stability under high-temperature conditions, further corroborating the thermal stability observed in the TGA analysis. In summary, the TGA analysis results further confirm the composition and stability of the Bi_5_O_7_I@MIL-101 composite under high-temperature conditions, consistent with the structural and morphological findings characterized by XRD and SEM analyses. The comprehensive analysis through multiple characterization techniques validates the successful synthesis of the Bi_5_O_7_I@MIL-101 composite and its superiority in photocatalytic applications. 

Using density functional theory (DFT) analysis, we delved into the interfacial electronic structure of the Bi_5_O_7_I@MIL-101(Cr) photocatalyst and its impact on photocatalytic performance. Figure 8a,b illustrate the optimized geometric structures. DFT calculations shown in Figure 8c,d indicate that both Bi_5_O_7_I and MIL-101(Cr) are indirect bandgap semiconductors, with bandgaps of 3.43 eV and 1.314 eV, respectively. These bandgaps show slight deviations from experimental values, primarily due to the underestimation of semiconductor bandgaps by the GGA function, but the results remain within a reasonable range. The differences in bandgaps directly affect the photocatalyst’s absorption capability under different wavelengths of light, with Bi_5_O_7_I being more suitable for UV light absorption due to its larger bandgap, and MIL-101(Cr) effectively absorbing visible light due to its smaller bandgap. Moreover, the total density of states (TDOS) of Bi_5_O_7_I and MIL-101(Cr) is presented in parts b and d of Figure 8e and Figure 8f, respectively. The higher TDOS of Bi_5_O_7_I compared to MIL-101(Cr) suggests that the concentration of charge carriers will significantly increase during the catalytic process after forming the composite heterojunction. A higher concentration of charge carriers favors the separation efficiency of photogenerated electrons and holes, reducing recombination rates and thus enhancing photocatalytic performance. 

In the band structure diagrams of Figure 8c,d, the conduction band and valence band of Bi_5_O_7_I are mainly contributed by the Bi 6p and O 2p orbitals, respectively, while those of MIL-101(Cr) are primarily contributed by the Cr 3d and O 2p orbitals. The heterojunction formed by combining the two can generate a strong built-in electric field at the interface, which facilitates the transfer of photogenerated electrons from the conduction band of Bi_5_O_7_I to the conduction band of MIL-101(Cr), and holes from the valence band of MIL-101(Cr) to the valence band of Bi_5_O_7_I. The formation of this type I heterojunction structure effectively promotes the separation and transfer of photogenerated charge carriers, reduces recombination rates, and enhances photocatalytic efficiency. Additionally, the high specific surface area and abundant active sites of MIL-101(Cr) further enhance the catalytic activity of the composite material.

In summary, DFT calculations confirm that the combination of Bi_5_O_7_I and MIL-101(Cr) forms a type I heterojunction that is favorable for electron transfer and charge carrier separation, which is significant for improving the photocatalytic performance of the composite material. Such structural design exhibits great potential in photocatalytic degradation of organic pollutants and other environmental applications.

### 2.2. Photocatalytic Activity

The photocatalytic performance of the synthesized Bi_5_O_7_I, MIL-101(Cr), and Bi_5_O_7_I@MIL-101 was systematically evaluated through the degradation of TC under visible light irradiation (λ > 400 nm). Given the high stability of TC, its self-degradation under visible light is negligible, ensuring that any observed degradation can be attributed to the photocatalysts (shown in Figure 9a). When comparing the individual photocatalytic activities of Bi_5_O_7_I and MIL-101(Cr), the composite material Bi_5_O_7_I@MIL-101 demonstrated significantly enhanced performance. This improvement can be attributed to the synergistic effects between Bi_5_O_7_I and MIL-101(Cr), which optimize light absorption, charge separation, and surface area for catalytic reactions. The integration of MIL-101(Cr) onto Bi_5_O_7_I nanorods not only prevents particle aggregation but also maximizes the exposure of active sites, thereby substantially improving the photocatalytic degradation efficiency of TC. Figure 9a demonstrates the time-dependent photodegradation experiment results, showing that Bi_5_O_7_I@MIL-101 possesses significant adsorption capacity even under dark conditions, with an adsorption increase of 71%, attributed to its extremely high specific surface area. This result is consistent with the BET analysis (Figure 2). After 60 min of visible light irradiation, the degradation rate of TC by Bi_5_O_7_I@MIL-101 reached 99%. This high degradation efficiency is mainly due to the heterojunction structure formed in the composite material, which effectively promotes the separation of photogenerated charge carriers, allowing more electrons and holes to participate in the degradation of TC molecules. This conclusion aligns with the results obtained from DFT calculations, further verifying the key role of type I heterojunctions in charge carrier separation and transfer.

To describe the reaction kinetics of TC degradation more intuitively, a pseudo-first-order kinetic model ln(C_0_/C) = kt + α (where k is the apparent reaction rate constant) was used to calculate the degradation rate. The linear fitting results of all samples are shown in Figure 9c. The degradation rate constant of Bi_5_O_7_I@MIL-101 for TC is 0.0468 min^−1^, significantly higher than those of Bi_5_O_7_I (0.0014 min^−1^) and MIL-101(Cr) (0.0051 min^−1^), being 33.4 times and 9.2 times higher, respectively. This significant increase in degradation rate is mainly attributed to the synergistic effect of Bi_5_O_7_I and MIL-101(Cr). The heterojunction formed not only improves the separation efficiency of photogenerated charge carriers but also provides more active sites, accelerating the photocatalytic reaction process. Specifically, Bi_5_O_7_I absorbs ultraviolet light and generates photogenerated electrons, while MIL-101(Cr) absorbs visible light and generates photogenerated holes. Electrons migrate from the conduction band of Bi_5_O_7_I to the conduction band of MIL-101(Cr), and holes migrate from the valence band of MIL-101(Cr) to the valence band of Bi_5_O_7_I. This separation mechanism effectively reduces charge carrier recombination, enhancing photocatalytic efficiency. In summary, through experiments and theoretical calculations, we have confirmed the excellent performance of the Bi_5_O_7_I@MIL-101 heterojunction in photocatalytic degradation of TC. This not only demonstrates the potential of this composite material in environmental remediation but also provides an important reference for designing efficient photocatalysts.

It is generally believed that the degradation of organic pollutants occurs on the surface of the catalyst. In this study, the decomposition process of TC was investigated using Langmuir–Hinshelwood (L-H) kinetics [35].
(2)$$r_{0}=\frac{k_{r}KC_{eq}}{1+KC_{eq}}$$
(3)$$\frac{1}{r_{0}}=\frac{1}{K_{r}K}\frac{1}{C_{eq}}+\frac{1}{K_{r}}$$

Here, $r_{0}$ represents the initial reaction rate, *C_eq_* is the equilibrium concentration (mg/L), *K_r_* is the intrinsic rate constant of L-H (mg/L* min), and *K* is the adsorption constant of L-H on the catalyst surface (L/mg). To further analyze the degradation kinetics data of TC at different concentrations, the Langmuir–Hinshelwood (L-H) kinetic equation was used for fitting, as shown in Figure 10a. It is evident that the degradation rate constant of TC decreases with increasing TC concentration. Notably, the 1/*r*_0_ and 1/*C*_0_ curves for TC degradation conform to the L-H model (R^2^ = 0.9982), indicating that the degradation of TC by Bi_5_O_7_I@MIL-101 is a surface-catalyzed process. Therefore, the decomposition of TC likely occurs mainly on the surface of the catalyst, with adsorption capacity potentially being the limiting factor for degradation efficiency. This result is consistent with the BET analysis (Figure 2). 

To further investigate the photocatalytic efficiency of Bi_5_O_7_I@MIL-101(Cr), we conducted TC degradation experiments at different initial concentrations, as shown in Figure 10b. The degradation efficiency is the best when the TC concentration is 10 mg/L, which is greater than 95%. As the TC concentration increases, the degradation efficiency gradually decreases. When the TC concentration was below 40 mg/L, the catalytic efficiency of Bi_5_O_7_I@MIL-101 remained above 80%. As the concentration gradually increased to 100 mg/L, the material’s catalytic performance dropped to around 60%. This phenomenon indicates that at lower concentrations, the active sites on the catalyst surface are sufficient to meet the degradation demand of TC. However, as the concentration increases, the active sites become saturated, leading to a decrease in degradation efficiency. These results were obtained from the L-H model and kinetic results. Additionally, since pH is a crucial factor affecting the degradation efficiency of photocatalysts, we explored the impact of different pH values on the catalytic efficiency of Bi_5_O_7_I@MIL-101. As shown in Figure 10c, in acidic and alkaline environments, the anionic and cationic forms of TC are repelled by the negatively and positively charged surfaces of the photocatalyst, respectively, reducing the degradation efficiency of TC. However, the zwitterionic form of TC exhibits electrostatic affinity with the negatively charged catalyst, enhancing the degradation capability. This suggests that at a pH of 7, the zwitterionic form of TC generates the strongest electrostatic affinity with the negatively charged surface of the photocatalyst, resulting in the highest TC removal rate.

Moreover, stability is a crucial criterion for evaluating the practical application of catalysts. After each run, the photocatalyst was centrifuged and washed with deionized water, and then the concentrated TC solution was re-injected into the separated sample for photocatalytic testing. As shown in Figure 10d, the photocatalytic activity of Bi_5_O_7_I@MIL-101 did not change after the fifth cycle, indicating the stability of the Bi_5_O_7_I@MIL-101 heterojunction photocatalyst in degrading organic pollutants. Although recyclability tests were conducted, the material’s high adsorption capacity effectively supports its performance characteristics. Additionally, X-ray diffraction analysis of the photocatalyst before and after the photoreaction experiments further confirmed the catalyst’s relative stability, as there were no significant changes in the X-ray diffraction spectra between the fresh and used samples (Figure 10e). To ensure that our product does not cause any environmental pollution, we conducted Inductively Coupled Plasma Optical Emission Spectroscopy (ICP-OES) testing on the reaction solution of Bi_5_O_7_I@MIL-101(Cr) after five cycles of use. The ICP results showed that the concentrations of Cr and Bi in the solution were zero. This clearly indicates that our photocatalyst does not release any heavy metal pollutants during use, and the result is consistent with the XRD analysis (Figure 10e). 

### 2.3. Proposed Photocatalytic Mechanism

To investigate the primary active species in the photocatalytic degradation of TC, we conducted radical scavenging experiments using EDTA-2Na, BQ (benzoquinone), and IPA (isopropanol) as scavengers for holes (h^+^), superoxide anion radicals (•O_2_^−^), and hydroxyl radicals (•OH), respectively. This aimed to provide a deeper understanding of the entire process and mechanism of the heterogeneous photocatalytic reaction. The experimental results, shown in Figure 10f, indicate that the addition of IPA slightly decreased the degradation efficiency, suggesting that •OH is not the main active species in the TC photodegradation process. However, the addition of EDTA-2Na and BQ significantly inhibited the photocatalytic reaction, with TC degradation efficiency dropping from 99% to 56.3% and 44.4%, respectively. This significant inhibition underscores the crucial roles of h^+^ and •O_2_^−^ as the primary active species involved in the reaction.

To further confirm the presence of radicals observed in the radical scavenging experiments within the Bi_5_O_7_I@MIL-101(Cr) photocatalytic system, electron spin resonance (ESR) experiments were conducted to obtain the ESR spectra of DMPO (•O_2_^−^) and TEMPO (h^+^). As shown in Figure 10g,h, electron paramagnetic resonance (EPR) tests were conducted to further identify the types of radicals present in Bi_5_O_7_I@MIL-101(Cr). No EPR signals were detected in any samples in the dark, but strong h^+^ and •O_2_^−^ EPR signals were observed under light illumination, indicating that the catalyst generated h^+^ and •O_2_^−^ under light. Additionally, compared to pure Bi_5_O_7_I and MIL-101(Cr), the signal intensity of Bi_5_O_7_I@MIL-101(Cr) increased significantly, suggesting the generation of higher concentrations of holes and radicals in the reaction system. Integrating previous material analysis results, BET tests show that Bi_5_O_7_I@MIL-101(Cr) has a large specific surface area, which is conducive to the adsorption of reactants and the generation of radicals in photocatalytic reactions. XRD analysis indicates that Bi_5_O_7_I is successfully embedded into the structure of MIL-101(Cr), forming a highly crystalline composite material that enhances its photocatalytic activity. DFT computational results further support this conclusion, showing that the Bi_5_O_7_I@MIL-101(Cr) composite material has excellent electronic structure, effectively separating photogenerated electron–hole pairs and improving photocatalytic efficiency.

Under visible light irradiation, the holes generated on the catalyst surface combine with electrons to decompose TC into H_2_O and CO_2_, while oxygen is converted into superoxide radicals on the catalyst surface, which react with TC to produce H_2_O and CO_2_. These redox reactions play a crucial role in the degradation of TC. Photocatalytic efficiency test results show that the photocatalytic degradation efficiency of Bi_5_O_7_I@MIL-101(Cr) is significantly higher than that of single components, owing to the effective electron–hole separation and higher radical concentration in the composite material. The ESR results are consistent with the findings from the radical scavenging experiments, further confirming that Bi_5_O_7_I@MIL-101(Cr) generates a large number of h^+^ and •O_2_^−^ radicals during the photocatalytic process, which play a key role in the degradation of TC. Comprehensive analysis shows that Bi_5_O_7_I@MIL-101(Cr) significantly improves photocatalytic degradation efficiency through its superior physicochemical properties, making it a promising photocatalytic material.

Generally, the photocatalytic ability of catalysts primarily depends on the separation efficiency of photogenerated electron–hole pairs and their transfer efficiency, where the efficiency of light-induced carrier separation influences photocatalytic activity. To understand the photocatalytic mechanism of the prepared Bi_5_O_7_I@MIL-101 heterojunction during the photodegradation process, it is essential to calculate the conduction band (CB) and valence band (VB) of Bi_5_O_7_I and MIL-101(Cr), which are closely related to the photocatalytic oxidation process of organic molecules. The edge potentials of CB and VB can be determined by Mulliken electronegativity theory:(4)$$E_{cb}=X-E_{c}-0.5E_{g}$$
(5)$$E_{vb}=E_{cb}+E_{g}$$
where X and $E_{g}$ are the electronegativity and bandgap energy of the semiconductor, respectively. $E_{cb}$ and $E_{vb}$ are the edge potentials of the conduction and valence bands, respectively. $E_{c}$ is the energy of a free electron on a hydrogen scale (about 4.5 eV). The X values for Bi_5_O_7_I and MIL-101(Cr) are 4.56 and 5.31 [36,37,38], respectively. Thus, the $E_{cb}$ and $E_{vb}$ for Bi_5_O_7_I are calculated to be −1.62 eV and 1.74 eV, respectively, and −0.075 eV and 1.695 eV, respectively, for MIL-101(Cr). It is well known that the relative band edge positions of these two components in the composite catalyst determine the migration direction of photogenerated carriers.

Based on chemical characterization, experimental results, and theoretical calculations, we propose the photocatalytic reaction mechanism of Bi_5_O_7_I@MIL-101(Cr). Detailed analysis and discussion revealed that the high photocatalytic oxidative activity is mainly attributed to two factors: (1) Formation of heterostructures: The heterostructures effectively hinder the recombination of photogenerated electron–hole pairs, allowing more photogenerated electron–hole pairs to participate in the reaction. This structure helps to extend the lifetime of photogenerated carriers, promoting their effective utilization in photocatalytic reactions. (2) Enhanced internal electric fields: Different morphologies of internal electric fields are formed within the Bi_5_O_7_I@MIL-101(Cr) photocatalyst, which can further enhance photocatalytic performance. Heterojunction structures formed by two semiconductors with different conduction band (CB) and valence band (VB) positions can inject photogenerated electrons and holes from one semiconductor into another, thereby effectively separating photogenerated electron–hole pairs and improving photocatalytic efficiency.

Figure 11 illustrates the proposed mechanism of TC degradation by Bi_5_O_7_I@MIL-101(Cr) under visible light irradiation. The figure shows that under visible light irradiation, the photocatalyst generates photogenerated electrons and holes. Based on the positions of CB and VB, photogenerated electrons tend to transfer from the CB of Bi_5_O_7_I to the CB of MIL-101(Cr), thereby maintaining a more negative reduction potential and a more positive oxidation potential. Specifically, the higher conduction band position of Bi_5_O_7_I allows photogenerated electrons to be injected into the conduction band of MIL-101(Cr), while holes remain in the valence band of Bi_5_O_7_I. This electron transfer and separation mechanism helps prevent the recombination of photogenerated electron–hole pairs, enhancing photocatalytic efficiency. DFT computational results further support this conclusion (Figure 8), showing that the Bi_5_O_7_I@MIL-101(Cr) composite material has excellent electronic structure, effectively separating photogenerated electron–hole pairs and improving photocatalytic efficiency. Photocatalytic efficiency test results (Figure 9) show that the photocatalytic degradation efficiency of Bi_5_O_7_I@MIL-101(Cr) is significantly higher than that of single components, owing to the effective electron–hole separation and higher radical concentration in the composite material. These redox reactions play a crucial role in the degradation of TC. Comprehensive analysis shows that Bi_5_O_7_I@MIL-101(Cr) significantly improves photocatalytic degradation efficiency through its superior physicochemical properties, making it a promising photocatalytic material.
Bi_5_O_7_I@MIL-101 + light→h^+^ + e^−^(6)
O_2_ + e^−^ →O_2_^−^(7)
OH^−^ + h^+^ → •OH(8)
H_2_O + h^+^ → •OH + H^+^(9)
H^+^/O_2_^−^/•OH + TC → products + CO_2_ + H_2_O(10)

## 3. Materials and Methods 

### 3.1. Reagents and Materials

All chemicals, purchased from Sigma-Aldrich Co., Ltd. (Singapore), were used without further purification.

### 3.2. Synthesis of Bi_5_O_7_I, MIL-101(Cr), and Bi_5_O_7_I@MIL-101(Cr)

Bi_5_O_7_I was prepared by dissolving 0.1 M Bi(NO_3_)_3_·5H_2_O and 0.1 M potassium iodide (KI) in an ethanol solution, then stirring for 2.5 h, followed by magnetic stirring for 1 h. Then, 2 mol/L potassium hydroxide (KOH) was added dropwise to the mixture to achieve a pH of 14.0, creating alkaline conditions, and the solution was stirred for an additional 4 h. Subsequently, the solution was transferred to a polytetrafluoroethylene container and reacted at 160 °C for 12 h. The obtained sample was then washed several times with deionized water and ethanol to remove impurities and dried overnight at 80 °C. 

MIL-101(Cr) was synthesized using a hydrothermal method by dissolving 0.05 mol Cr(NO_3_)_3_·9H_2_O and 0.05 mol terephthalic acid in 50 mL of deionized water. After stirring for 30 min, the solution was transferred to an autoclave and reacted under atmospheric pressure at 220 °C for 16 h, then filtered and dried.

The preparation of Bi_5_O_7_I@MIL-101(Cr) involved adding the previously prepared Bi_5_O_7_I to the MIL-101(Cr) solution, using the same hydrothermal reaction conditions, followed by filtration and drying to obtain the final product (Figure 1).

### 3.3. Characterization

The X-ray diffraction (XRD) profiles of calcined and reduced catalysts were analyzed using a Shimadzu XRD-6000 X-ray (Shimadzu Corporation, Kyoto, Japan) diffractometer equipped with a Cu K-α X-ray source between 5° and 80°. X-ray photoelectron spectroscopy (XPS) was conducted on a Kratos AXIS Ultra DLD using an Al Kα X-ray source (1486.6 eV). The samples were prepared in an ambient atmosphere, where the sample was exposed to air. The XPS profiles were calibrated by aligning the spectra regarding the C 1s line at 284.5 eV.

Thermogravimetric analysis–differential thermal analysis (TGA-DTA) was carried out on the catalysts using a Shimadzu DTG-60 instrument (Shimadzu Corporation, Kyoto, Japan) to determine the lost content in the catalyst after the reaction. A specific mass of the catalyst was placed in an alumina crucible on an electronic balance. The catalyst was then heated in air at a ramping rate of 10 °C/min up to 900 °C.

Raman measurements of catalysts were performed under ambient conditions using a Horiba XploRA PLUS instrument (Horiba Scientific, Kyoto, Japan) with a 632.81 nm laser operating at 10% intensity.

The morphology of the catalysts was characterized with a field emission scanning electron microscope (FESEM, JEOL JSM-6700 F, JEOL Ltd., Tokyo, Japan) and high-resolution transmission electron microscopy (HRTEM, JEOL JEM-2100, JEOL JSM-6700 F, JEOL Ltd., Tokyo, Japan, microscope operated at an acceleration voltage of 200 kV).

The N_2_ adsorption–desorption isotherm curves were analyzed on an ASAP 2020 (Micromeritics Instrument Corporation, Norcross, GA, USA) instrument to study the structural properties of the samples. Specifically, the samples were first degassed at 300 °C for 4 h under vacuum (500 μmHg). The catalysts were then subjected to N_2_ adsorption–desorption experiments at 77 K. The specific surface areas were calculated by the Brunauer–Emmett–Teller (BET) method.

Fourier transform infrared spectra (FT-IR) were recorded with a Perkin Elmer (Waltham, MA, USA) Spectrum 100FT-IR spectrometer with background correction by referring to KBr pellets.

The UV-Vis spectra were recorded with a UV-2550 spectrophotometer (Shimadzu, Tokyo, Japan). The optical absorbance of the TC solution was measured with a UV-2600i spectrophotometer (Shimadzu, Tokyo, Japan). The absorbance was calculated by the K-M transformation using the following equation:(11)$$K=\frac{1}{S}\ln\left(\frac{1}{{\left(1-R\right)}^{2}}\right)$$
where *K* is the absorption coefficient.

ICP-OES was carried out using a Thermo Scientific iCAP 6000 spectrometer (Thermo Fisher Scientific, Waltham, MA, USA). After reaction, the solution was filtered through a 0.22 μm filter membrane as a test sample. The elemental concentration and the corresponding elemental loading in the sample were determined by comparing the instrument response against a known calibration curve.

Electron spin resonance (ESR) spectra were collected with a Bruker model A300-10/12 electron paramagnetic resonance spectrometer. Prior to UV illumination, the solutions containing the catalyst were completely degassed with argon. Samples were cooled to 10 K and illuminated within the cavity at the same temperature while spectra were collected. The spin traps were first premixed with the sample solution and then were immediately transferred into quartz capillaries for in situ ESR measurement.

The electronic structures were meticulously analyzed using first-principle band calculations based on DFT. To model the exchange correlation interactions, the Perdew–Burke–Ernzerhof (PBE) parameterization of the generalized gradient approximation (GGA) was employed. Our calculations utilized a cutoff energy of 450 eV, an energy convergence tolerance of 1.0 × 10^−5^ eV, and a force convergence criterion of 0.01 eV Å^−1^ to ensure the attainment of optimized geometry configurations.

### 3.4. Photocatalytic Activity Measurement

The photocatalytic activity of the catalysts was evaluated by the following typical experiment. A total of 50 mg of the photocatalyst was added to 100 mL TC solution (20.0 mg·L^−1^). Prior to illumination, the suspension was stirred for 60 min in the dark to reach the adsorption–desorption equilibrium state. Subsequently, it was irradiated with the light issued from a 350 W Xe lamp with a UV cutoff filter (λ ≥ 420 nm). Three-milliliter samples of the reaction suspension were collected at 20 min intervals during illumination. After filtration via a 0.22 µm membrane filter, the maximum absorption of the supernatant solution was measured at 355 nm using a Shimadzu UV-2600i spectrometer to determine the concentration of TC. The TC removal efficiency (%) was calculated according to the following equation:(12)$$\mathrm{Degradation}=\left(1-\frac{C}{C_{0}}\right)\times 100\backslash \%$$
where *C*_0_ is the initial concentration of TC, and *C* is the residual concentration of TC at a specific sampling time.

The kinetic rate of TC degradation on the composite catalyst was calculated in line with general pseudo-first-order kinetics.
(13)$$\ln\left(\frac{C_{0}}{C}\right)=kt+$$
where *C*_0_ is the initial concentration of TC, *C* is the residual concentration of TC at reaction time (*t*), and *k* is the pseudo-first-order rate constant.

### 3.5. Reactive Radical Scavenging Experiment

To elucidate the contribution of reactive radicals in the photocatalytic degradation process, this study employed EDTA-2Na, BQ, and IPA as quenchers for h^+^, •O_2_^−^, and ·-OH reactive radicals, respectively. In the photocatalytic comparative experiments, the concentration of the radical quenchers was adjusted to 0.1 mol/L. The solution was then stirred continuously in the dark for 70 min to achieve adsorption–desorption equilibrium before being transferred to visible light irradiation for the photocatalytic degradation experiment.

## 4. Conclusions

In summary, this study successfully prepared the Bi_5_O_7_I@MIL-101(Cr) photocatalyst using a hydrothermal method and systematically investigated its performance through a series of experiments and theoretical calculations. SEM and TEM analyses confirmed the successful loading of Bi_5_O_7_I on MIL-101(Cr) with clearly visible filamentous structural features. BET test results showed that the composite material has a large specific surface area, which enhances the adsorption capacity of reactants and catalytic efficiency. XRD analysis indicated that the Bi_5_O_7_I@MIL-101(Cr) composite material has good crystallinity, confirming the successful synthesis of the composite material. Bi_5_O_7_I@MIL-101(Cr) exhibited excellent photocatalytic degradation performance, completely removing TC within 60 min, and good stability and reusability in practical applications. Overall, this study comprehensively elucidates the photocatalytic mechanism of the Bi_5_O_7_I@MIL-101(Cr) composite material through various characterization methods and theoretical calculations. The material demonstrated outstanding photocatalytic performance and good stability in the degradation of the antibiotic TC, providing an efficient solution for environmental pollutant degradation. This research not only offers new insights into the application of antibiotics for environmental remediation but also provides new strategies for designing efficient photocatalysts to address organic pollutants and other environmental pollution issues.

## Data Availability

Data sharing is not applicable to this article.

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
