# Peer review of "Innovative Bi5O7I/MIL-101(Cr) Compounds: A Leap Forward in Photocatalytic Tetracycline Removal"

_ijms, 2024, doi:10.3390/ijms25126759_

Round 1

Reviewer 1 Report (Previous Reviewer 2)

Comments and Suggestions for Authors

The Authors have carefully reviewed the entire manuscript  “Innovative Bi5O7I/ MIL-101(Cr) Compounds: A Leap Forward in Photocatalytic Tetracycline Removal” responding, point by point, to the suggestions propose in the first review process. They have corrected some typos and errors, making the text more fluid and clearer for the reader.

The new version of the paper presents additional information on the characterization of the studied photocatalysts, such as its porose structure, its adsorption capacity, and its chemical structure. The Authors have additionally conducted leaching tests by means of ICP-OES measurements demonstrate that the proposed catalyst is free from heavy metal contaminations.

It was also investigated the decomposition process of tetracycline using the Langmuir-Hinshelwood (L-H) kinetics model demonstrating that the degradation of pollutant occurs predominantly on the catalyst surface.

Furthermore, an accurate analysis of the possible degradation processes that could be involved was proposed, by means of additional radical scavenging experiments.

Therefore, considering the corrections made, the better interpretation of the experimental data, and the new measurements carried out, I believe that the revised version of the manuscript presents a good scientific level and a high accuracy of data such that it can be published in the International Journal of Molecular Sciences.

Author Response

Thank you very much for taking the time to review this manuscript. We have revised the manuscript to ensure a more fluid and clear text. Thank you for your feedback. These additions and modifications enhance the robustness of our study and address the practical implications of deploying this photocatalyst in real-world settings. Please find the detailed responses below and the corresponding revisions in the submitted files.

Reviewer 2 Report (New Reviewer)

Comments and Suggestions for Authors

The paper of Hong et al. relates about the development of Bi5O7I/ MIL-101(Cr) compounds for TC degradation. It is well constructed and discussed, since it is rich of characterization results and photocatalytic tests. So, it is suitable for the publication after some minor revisions:

1- regarding the antibiotics’ introduction, the ref.s https://doi.org/10.1021/acsphotonics.3c0072, https://doi.org/10.1016/j.scitotenv.2024.171757, https://doi.org/10.1080/10643389.2020.1859289 should be added (line 46). 

2- Surface area values have to be indicated with full numbers (not commar) due to the error associated to the type of measurement.

3- Line 57: BiOx, where X must be amended in uppercase.

4- in my opinion, point of zero charge determination is a characterization technique that should be added to predict the adsorption capability of the materials (according to the ref. cited in 1).

5- Table 1 can be deleted, since the result is clearly explained in the main text.

6-photolysis of TC should be done and added.

7- English lexicon can be improved.

Comments on the Quality of English Language

English lexicon can be improved.

Author Response

Thank you very much for taking the time to review this manuscript. We have revised the manuscript to ensure a more fluid and clear text. Thank you for your feedback. These additions and modifications enhance the robustness of our study and address the practical implications of deploying this photocatalyst in real-world settings. Please find the detailed responses below and the corresponding revisions in the submitted files.

Reviewer 3 Report (New Reviewer)

Comments and Suggestions for Authors

The article “Innovative Bi5O7I/ MIL-101(Cr) Compounds: A Leap Forward in Photocatalytic Tetracycline Removal” submitted to IJMS by Hong et al. presents results on the synthesis, characterisation and photocatalytic properties of a a bismuth oxyiodide / MIL-101(Cr) metal organic framework composite for the removal of the antibiotic tetracycline.

Starting with a positive impression - the authors have really shown lots of effort to conduct the study - it includes almost any possible analysis for characterisation, including XRD, BET, SEM-EDX, TEM-EDX, XPS, FTIR, Raman, UV/Vis DRS, TGA, and even DFT calculations, + EPR. The thoroughness of the photocatalytic properties investigation is also impressive - the authors conducted kinetic analysis & dark-adsorption analysis of TC removal, cycling, effects of pH, effects of TC concentration, effects of active photogenerated species scavengers and commented on a mechanism. The amount of work is impressive. The quality of English (at least in terms of grammar and punctuation) is good. 

However, I cannot recommend the work for publication. I am absolutely certain that the amount of data and results shown by the authors could yield an amazingly good work, so I hope that they are not discouraged, but the level of revision required in order to convert the text into a publishable work is beyond major. It needs a complete re-writing, and the discussion, which is seriously lacking, needs to be improved, because in its current state is quite worrisome.

Here are some hints and suggestions:

(1) I suspect that during past revisions new analyses were added, however, they are not properly implemented in the text, nor described: e.g., the EPR analysis (DMPO / TEMPO trapping experiments) is not mentioned anywhere in the materials and methods section; The leaching tests were not described as well, even though the ICP-OES is mentioned in the experimental section, it states that samples were digested in HF and analysed (which ? The catalysts) - according to table 1 this should imply that they have 0 wt.% Cr, Bi, and I, so one would guess not, but the text (line 255-256) states “… our catalyst is free from any heavy metal contamination.”. The authors need to describe the procedure thoroughly and logically it’s completely unclear here what was leached and what was analysed and quite confusing; (3) this also includes the DFT part - nothing is mentioned about which software was used, which functional, etc.; additionally due to additions and re-arrangements some of the figures are misnumbered or not addressed in the text (e.g., Figure 9). Additionally, some of the figures are with illegible quality - the SEM and TEM images in Figure 1 don’t give any sense of scale, since the scale markers are not visible; 

(2) What is more worrying is, however, the lack of quality in the discussion of the results. Many of these multiple analyses are discussed almost ad-hoc with not much effort and - in many cases erroneously. An example is the DFT results, where it is stated that “both Bi5O7I and MIL-101(Cr) are indirect bandgap semiconductors” (Line 276), but then the Tauc analysis (line 230-231) states “The value of (n) depends on the transition characteristics of the semiconductor, which in this case is 2”, which is not true and in the Tauc plot (αhν)² is used for a direct bandgap semiconductor, while 1/2 (square root of (αhν)) is used for indirect bandgap semiconductors. The treatment of the DRS is also strange, since it shows “Absorbance” on Figure 6, but the authors do not suggest that any K-M transportation was applied.

(3) But Point (2) is relatively minor, since the Tauc plots are routinely used, however, routinely applied wrongly in the field of photocatalysis - a bigger issue is also the entire paragraph in lines 188 - 200 talking about D and G bands. In lines 185-186 the authors correctly assign the bands at 1400 and 1625 cm-1 as ones belonging to the organic linker in the MOF, and obviously there are no bands at 1350 cm-1 nor 1593 cm-1. D & G notation is observed in 2D carbon-based materials (graphene, graphene oxide) but why are the authors discussing this given that (i) there is no 2D graphitic material in their composite and (ii) that obviously these bands are not visible on the Raman spectrum, and they even managed to calculated Id/Ig ratio ? And sadly on many places in the text it is like this “, suggesting that the formation of defect characteristics. From Raman spectra, we can infer that ….may exhibit enhanced interactions and or synergistic effects, … related to different structural units…. potentially affect photocatalytic activity, … electron-hole pair separation”… etc. This entire paragraph in lines 195-200 means absolutely nothing. It is just a vague statement, that based on this analysis we found that there is heterojunction (and pretty much the same discussion is used in explaining the TEM images, the UV-vis absorbance etc.

(4) However, points 1-3, while an example of things that could be improved via major revision, there is one very noticeable error in the data treatment, that I am not sure how got missed, but a revision required to rectify it would completely change the interpretation of the photocatalytic results, along the conclusions and main claims in the abstract. It is absolutely obvious in Figure 9, that when the kinetic constants and degradation efficiencies were calculated via the C/C0 dependence - the authors used the initial absorbance before the dark phase (and judging by difference between the MIL-101 and Bi5O7I@MIL-101 in Figure 9b, possibly, somehow selectively - only in the later case)! This is simply wrong and makes no sense. The entire purpose of using the -ln(C/C0) approach is, that the concentration of the model contaminant does marginally not affect the kinetic constants (where in fact it does). The data in Figure 9 seems very odd, since it is obviously that there is a miscalculation and that the composite is only marginally more active than the MOF (possibly twice, but not 7.5 times, and I also suspect not 17 times more active than the pristine Bi5O7I, which is stated in the abstract). This may also be translated to the data in Figure 10d. 

So, ultimately I sense that the paper is flawed and should not be considered for publication in IJMS and be rejected with an encouragement for a resubmission. The authors should not be discouraged by this, but should really take time to carefully re-analyse their data and then re-structure their manuscript.  

Author Response

Thank you very much for taking the time to review this manuscript. We have revised the manuscript to ensure a more fluid and clear text. Thank you for your feedback. These additions and modifications enhance the robustness of our study and address the practical implications of deploying this photocatalyst in real-world settings. Please find the detailed responses below and the corresponding revisions in the submitted files.

Round 2

Reviewer 3 Report (New Reviewer)

Comments and Suggestions for Authors

I would like to thank the authors for the response to the reviewers' suggestions and their effort to improve the manuscript. 

I am pleased with the re-calculation of the photocatalytic oxidation (PCO) performance data, which is now correctly calculated and presented. In this respect - would also like to acknowledge that while my initial suggestion that the manuscript is rejected due to the poor and erroneous processing of the PCO data, actually my harsh opinion that the Bi5O7I/MIL-101 composite mix is no more than 2x more active than the two components of the mix was erroneous as well (since, of course it having a higher surface area and a higher saturation coverage, leading to a much lower initial concentration during the "light" phase, means that that it degrades up to 100% of a lower total TC concentration - this is an issue with these C/C0 kinetic analyses in general, however, I am certain that the overall calculation is done correctly by the authors now). So, I would appologise on my behalf and congratulate the authors that their composite now is deemed even more reactive than the original manuscript :).

Finally, I would endorse the manuscript for publication now, after the revision, and would like to wish the authors even more fruitful future works. 

This manuscript is a resubmission of an earlier submission. The following is a list of the peer review reports and author responses from that submission.

Round 1

Reviewer 1 Report

Comments and Suggestions for Authors

The manuscript presents an interesting heterojunction but is affected by some problems.

There are several typos.

The caption of fig.4 should be changed.

The high SSA of composite photocatalyst should be differently evaluated. BET model fails with microporous system, and the N2 adsorption -desorption isotherm is very strange. The attribution to IUPAC classification should be firstly effected, then the right model should be chosen for the SSA or porosity evaluation. 

However, the high SSA should justify the high ability in the adsorption of tetracycline, which is shown in Fig. 9a, where is clear that the dark  adsorption is not ended. As confirmation, the pseudo-first kinetic model fails to be correct, and the authors have applied it without criticism. Indeed the intercepts should be zero!!!! Only for BiOI seems ok.

Experiments by varying the initial concentration of tetracycline should be performed, as well as by changing pH. 

Furthermore, the dark adsorption should be more carefully studied, also considering a Languimir Hinshelwood model.

Even if the recyclability tests were performed, a high adsorption capacity could justify the behaviour. 

Cr leaching tests should be added , as well as verification of TOC removal.

Comments on the Quality of English Language

the English is acceptable

Author Response

Response to Reviewer 1 Comments

1. Summary

Thank you very much for taking the time to review this manuscript. These additions and modifications enhance the robustness of our study and address the practical implications of deploying this photocatalyst in real-world settings. Please find the detailed responses below and the corresponding revisions in the re-submitted files.

2. Questions for General Evaluation

Reviewer’s Evaluation

Response and Revisions

Does the introduction provide sufficient background and include all relevant references?

Yes

Are all the cited references relevant to the research?

Can be improved

Is the research design appropriate?

Can be improved

Are the methods adequately described?

Yes

Are the results clearly presented?

Must be improved

Are the conclusions supported by the results?

Must be improved

3. Point-by-point response to Comments and Suggestions for Authors

Comments 1: There are several typos.

Response 1: Thank you for pointing this out. We have carefully checked the text and corrected any typos.

Comments 2: The caption of fig.4 should be changed.

Response 2: Thanks for clarifying that for us. We have corrected the caption of Fig.4.

Comments 3: The high SSA of composite photocatalyst should be differently evaluated. BET model fails with microporous system, and the N2 adsorption -desorption isotherm is very strange. The attribution to IUPAC classification should be firstly effected, then the right model should be chosen for the SSA or porosity evaluation.

Response 3: We are grateful for your detailed explanation. According to the suggestions of Reviewer 1, we have repeated the experiment several times and made some corrections to the data. We have updated the figure and analysis in pages 4 and 5.

Comments 4: However, the high SSA should justify the high ability in the adsorption of tetracycline, which is shown in Fig. 9a, where is clear that the dark  adsorption is not ended. As confirmation, the pseudo-first kinetic model fails to be correct, and the authors have applied it without criticism. Indeed the intercepts should be zero!!!! Only for BiOI seems ok.

Response 4: Thank you for pointing this out. We have repeated the experiment and extended the time up to adsorption saturation. The new result is presented in page 11.

Comments 5: Experiments by varying the initial concentration of tetracycline should be performed, as well as by changing pH.

Response 5: Thanks for clarifying that for us. We have added the experiment of different pH and concentration in pages 10-12.

Comments 6: Furthermore, the dark adsorption should be more carefully studied, also considering a Languimir Hinshelwood model.

Response 6: We appreciate you sharing this information with us. We have added analysis and plotting of Langmuir- Hinshelwood model, with the details as shown in page 10-12.

Comments 7: Even if the recyclability tests were performed, a high adsorption capacity could justify the behaviour.

Response 7: We agree with Reviewer 1. We have incorporated this perspective into our analysis, which can be found on page 10.

Comments 8: Cr leaching tests should be added , as well as verification of TOC removal.

Response 8: Thank you for pointing this out. We have added the test in page 8 and 9.

Reviewer 2 Report

Comments and Suggestions for Authors

The manuscript  “Innovative Bi5O7I/ MIL-101(Cr) Compounds: A Leap Forward in Photocatalytic Tetracycline Removal” reports synthesis and characterization of a hybrid catalyst used to degrade tetracycline.

The paper reports a complete morphological and structural characterization of the synthesized materials, in addition a detailed study of its photocatalytic performance was performed and described.

In my opinion, the manuscript presents a good interpretation of scientific data, however there are some imprecisions, and the following concerns should be addressed in order to improve the paper quality and increase the reader interest:

- Please correct some typos present in the text.

- Numerous papers report the use of photocatalysts with heterojunctions for degradation of tetracycline (for example doi.org/10.3390/catal14030174, doi.org/10.3390/catal11101238, doi.org/10.3389/fchem.2022.1069816). In my opinion, the authors should explain the novelty of their proposed research compared to other work present in the literature.

- Please check the correct names of imagines in Figure 1. Some inaccuracies appear both in the test (paragraph 2.1) and in the caption of Figure 1.

- Caption of Figure 2 should be also written better.

- The FTIR spectra are presented in Figure 4 and not in Figure 3 as reported in line 158.

- Comment of Raman spectra should be better contextualized (it is not immediately clear when the Authors start to analyze the Raman spectra).

- Caption of Figure 4 must be corrected.

- Do the Authors believe that the used catalysts could be reused for further degradation cycles?

Author Response

Response to Reviewer 2 Comments

1. Summary

Thank you for your insightful feedback and suggestions to enhance our manuscript. We believe these changes will satisfy the concerns raised and provide a more comprehensive understanding of our photocatalyst's performance. Please find the detailed responses below and the corresponding revisions in the re-submitted files.

2. Questions for General Evaluation

Reviewer’s Evaluation

Response and Revisions

Does the introduction provide sufficient background and include all relevant references?

Can be improved

Are all the cited references relevant to the research?

Can be improved

Is the research design appropriate?

Yes

Are the methods adequately described?

Can be improved

Are the results clearly presented?

Can be improved

Are the conclusions supported by the results?

Can be improved

3. Point-by-point response to Comments and Suggestions for Authors

Comments 1: Please correct some typos present in the text.

Response 1: Thank you for pointing this out. We have carefully checked the text and corrected all typos.

Comments 2: Numerous papers report the use of photocatalysts with heterojunctions for degradation of tetracycline (for example doi.org/10.3390/catal14030174, doi.org/10.3390/catal11101238, doi.org/10.3389/fchem.2022.1069816). In my opinion, the authors should explain the novelty of their proposed research compared to other work present in the literature.

Response 2: Thanks for clarifying that for us. We have explained the novelty of our proposed research compared to these works in the introduction part on page 2.

Comments 3: Please check the correct names of imagines in Figure 1. Some inaccuracies appear both in the test (paragraph 2.1) and in the caption of Figure 1.

Response 3: We are grateful for pointing this out. We have carefully checked the full text and revised it accordingly.

Comments 4: Caption of Figure 2 should be also written better.

Response 4: Thank you for pointing this out. We have carefully checked the full text and revised it accordingly.

Comments 5: The FTIR spectra are presented in Figure 4 and not in Figure 3 as reported in line 158.

Response 5: Thanks for clarifying that for us. We have carefully checked the full text and revised it accordingly.

Comments 6: Comment of Raman spectra should be better contextualized (it is not immediately clear when the Authors start to analyze the Raman spectra).

Response 6: I appreciate you sharing this information with us. We have reanalyzed the Raman spectrum, please refer to page 6-7 for further details.

Comments 7: Caption of Figure 4 must be corrected.

Response 7: Agree. We have modified it, please refer to page 7.

Comments 8: Do the Authors believe that the used catalysts could be reused for further degradation cycles?

Response 8: Thank you for pointing this out. Yes, we believe that spent catalyst can be reused for further degradation cycles. For this purpose, we also conducted relevant cyclic degradation tests. For details, please refer to Figure 10d.